# Decreases and Pronounced Geographic Variability in Antibiotic Prescribing in Medicaid

**DOI:** 10.3390/pharmacy12020046

**Published:** 2024-03-01

**Authors:** Alexia G. Aguilar, Priscilla C. Canals, Maria Tian, Kimberly A. Miller, Brian J. Piper

**Affiliations:** 1Department of Medical Education, Geisinger Commonwealth School of Medicine, Scranton, PA 17822, USA; aaguilar@som.geisinger.edu (A.G.A.); pcanals@som.geisinger.edu (P.C.C.); mtian@som.geisinger.edu (M.T.); kmiller36@geisinger.edu (K.A.M.); 2Center for Pharmacy Innovation and Outcomes, Danville, PA 17822, USA

**Keywords:** antimicrobial resistance, pharmacoepidemiology, prescribing, stewardship

## Abstract

Antibiotic resistance is a persistent and growing concern. Our objective was to analyze antibiotic prescribing in the United States (US) in the Medical Expenditure Panel System (MEPS) and to Medicaid patients. We obtained MEPS prescriptions for eight antibiotics from 2013 to 2020. We extracted prescribing rates per 1000 Medicaid enrollees for two years, 2018 and 2019, for four broad-spectrum (azithromycin, ciprofloxacin, levofloxacin, and moxifloxacin) and four narrow-spectrum (amoxicillin, cephalexin, doxycycline, and trimethoprim-sulfamethoxazole) antibiotics. Antibiotic prescriptions in MEPS decreased from 2013 to 2020 by 38.7%, with a larger decline for the broad (−53.7%) than narrow (−23.5%) spectrum antibiotics. Antibiotic prescriptions in Medicaid decreased by 6.7%. Amoxicillin was the predominant antibiotic, followed by azithromycin, cephalexin, trimethoprim-sulfamethoxazole, doxycycline, ciprofloxacin, levofloxacin, and moxifloxacin. Substantial geographic variation in prescribing existed, with a 2.8-fold difference between the highest (Kentucky = 855/1000) and lowest (Oregon = 299) states. The South prescribed 52.2% more antibiotics (580/1000) than the West (381/1000). There were significant correlations across states (r = 0.81 for azithromycin and amoxicillin). This study identified sizable disparities by geography in the prescribing rates of eight antibiotics with over three-fold state-level differences. Areas with high prescribing rates, particularly for outpatients, may benefit from stewardship programs to reduce potentially unnecessary prescribing.

## 1. Introduction

Antibiotic resistance has been deemed to be one of the most pressing threats to present-day public health. According to the World Health Organization, the rise in antibiotic resistance is one of the most concerning risks to global health, food security, and the development of human defenses [1]. Overuse and inappropriate prescribing of antibiotics have been identified in a variety of healthcare settings [2,3,4,5]. According to a 2019 report from the Centers for Disease Control and Prevention, more than 2.8 million antibiotic-resistant infections occur in the United States (US) each year, and more than 35,000 people die secondarily [6]. Most of these antibiotic prescriptions were for respiratory infections, commonly caused by viruses, which do not respond to antibiotics [7]. Antibiotics are often prescribed unnecessarily [8,9], and 30–50% of prescriptions for them were not associated with an indication [10,11,12]. Sulfonamides and urinary anti-infective agents are the classes most likely to be prescribed without a documented indication [11]. In one analysis, 25% of antibiotics prescribed in the outpatient setting to Medicaid beneficiaries were not associated with a provider visit, and therefore, were not screened by existing antimicrobial stewardship systems [13]. Further, among 298 million prescriptions filled by 53 million Medicaid patients between 2004 and 2013, 45% of the prescriptions for antibiotics were made without any clear rationale. Twenty-eight percent of antibiotics were prescribed without evidence of seeing the provider, and 17% were given without documentation for infection-related diagnosis. Inappropriate antibiotic prescribing leads to antibiotic resistance in an individual as well as at the community level, particularly against public health threats such as carbapenem-resistant *Enterobacteriaceae* and methicillin-resistant *Staphylococcus aureus* [14]. Furthermore, overuse of antibiotics increases the risk of adverse effects such as rash, GI upset, and renal dysfunction [6]. The risk of infection with *Clostridioides difficile* increases with duration of exposure to antibiotics [15,16].

State-level differences in antibiotic prescribing are well known [17,18]. Geographic areas with high antibiotic consumption have been associated with increased antibiotic resistance, and broad-spectrum antibiotics were shown to be more likely to be associated with antibiotic resistance. The South census region had the highest prescribing rate in the US [19,20]. Among commercially insured patients in 2017 with an acute respiratory tract infection, those living in the South were 34% more likely than those in the West to receive an antibiotic [20]. Similarly, examination of medical records from the National Ambulatory Medical Care Survey from 2012 to 2013 revealed that patients in the East South Central region (i.e., AL, KT, MS, TN) were more likely than those in Pacific states (CA, OR, WA) to be diagnosed with a condition where an antibiotic would be prescribed [21]. Kentucky had the highest prescription rates (1281 prescriptions per 1000 persons), which was about four-fold higher than Alaska (348 per 1000 persons) [17]. An analysis of nationally representative data from the Medical Expenditure Panel Survey (MEPS) [22] revealed relatively stable antibiotic use overall from 2000 (382/1000 persons) to 2010 (384/1000) but a doubling for broad-spectrum antibiotics [19]. Relative to 27 European countries, the US ranked fourth in 2004 for daily doses of antibacterials and was number one for azithromycin, levofloxacin, and cefdinir [23].

While regional and state-level disparities in antibiotic prescribing are well known [20,21], it is important to provide updated information to provide information on whether stewardship efforts have been successful in driving down these population-level differences. The low-income Medicaid population may face challenges that are different from those that are commercially insured. Due to the public health risks that arise from antibiotic resistance [1,6], and other adverse health outcomes associated with unnecessary exposure to antibiotics [24], it is imperative to continuously monitor antibiotic prescribing patterns to identify potentially unnecessary prescriptions and inform stewardship efforts. This study examined the temporal profile of outpatient antibiotic use as reported by MEPS and geographical patterns of antibiotic prescribing rates among US Medicaid program beneficiaries.

## 2. Materials and Methods

### 2.1. Participants

The MEPS is a national set of surveys of patients, families, and their medical providers with a sample of 35,000 respondents to estimate healthcare use annually which has been employed in prior research [20]. MEPS is sponsored by the Agency for Healthcare Research and Quality and is nationally representative of the US civilian non-institutionalized population. The pharmacy component involves verifying patient-reported information about prescription drugs [22]. Medicaid is a joint federal and state program that provides coverage for 75 million people or 21% of the US population [25]. It is one of the largest payers for healthcare in the US. All states provide coverage for outpatient prescription drugs [25].

### 2.2. Procedures

Antibiotic agents that were consistently ranked in the top three-hundred most prescribed antibiotics in MEPS were examined from 2013 to 2020. This period was selected to provide an update on past research [19]. These antibiotics included amoxicillin (including amoxicillin-clavulante), azithromycin, cephalexin, ciprofloxacin, doxycycline, levofloxacin, and trimethoprim-sulfamethoxazole. These antibiotics were examined from 2013 to 2020 (the most recent available when data analysis was completed in December 2022). The outcomes were the total annual number of prescriptions, broad vs. narrow spectrum, and number of prescriptions for each agent.

Quarterly state antibiotic prescription totals and population data during 2018–2019 were collected from the Medicaid State Drug Utilization database. These years were selected because of their relative recency as well as because of their being prior to disruptions associated with the COVID-19 pandemic. Antibiotics were identified using the generic and trade names [26]. Antibiotics were further categorized as broad- or narrow-spectrum based on their potential for influencing antibacterial resistance as well as their spectrum of activity according to the National Committee for Quality Assurance (NCQA) Antibiotics of Concern list. Azithromycin, levofloxacin, moxifloxacin, and ciprofloxacin were categorized as broad-spectrum antibiotics, while amoxicillin, cephalexin, doxycycline, and trimethoprim-sulfamethoxazole were categorized as narrow-spectrum antibiotics [27,28]. These agents were selected based on overlap with those listed in MEPS as well as past Medicaid investigations [17]. The states were divided into geographic regions according to the US Census (Appendix A). The primary outcome was the number of prescriptions (yearly and quarterly) corrected for the number of enrollees.

### 2.3. Data Analysis

The prescriptions as well as patients in MEPS were extracted. The overall total was calculated for each year and for broad- and narrow-spectrum agents. After the total number of prescriptions was collected and the prescription rate per 1000 Medicaid enrollees was calculated, we input the data into IMB SPSS. Pearson correlations were completed between state-level antibiotic use (prescriptions/1000 enrollees) for each year. A positive r value indicated that states that prescribed more of antibiotic A also prescribed more of antibiotic B. Although r > 0.279 was significant (*p* < 0.05) for associations with fifty states, correlations above 0.50 were considered large. We created figures and heatmaps through Statistical Analysis System’s John’s Macintosh Project (SAS JMP) and GraphPad Prism to show the variation across regions in the United States. Variability was reported as the standard error of the mean (SEM). A *p*-value of <0.05 was considered statistically significant.

## 3. Results

### 3.1. MEPS

During the study period (2013–2020) there were 687,979,538 million antibiotic prescriptions dispensed in the US. Total antibiotic prescriptions decreased by 38.7% while the US population increased by 5.2% from 2013 to 2020. Broad-spectrum agents had a larger decline (−53.7%) than narrow-spectrum agents (−23.5%). More specifically, amoxicillin prescriptions decreased by 39.1%, trimethoprim-sulfamethoxazole by 25.9%, and cephalexin by 8.9%, while there was increase in doxycycline prescriptions (+16.2%). Azithromycin (−68.3%), levofloxacin (−56.4%), and ciprofloxacin (−33.8%) prescriptions experienced pronounced declines while amoxicillin-clavulanate increased (+3.9%; Figure 1A). Similarly, the total number of patients that were dispensed antibiotics decreased from 71.3 million in 2013 to 42.7 million (−40.0%) in 2020. The reduction in recipients of broad-spectrum agents (−55.6%) was over twice as large as narrow-spectrum (−23.3%).

### 3.2. Medicaid

The total amount of prescriptions in 2019 (33,011,946 prescriptions) was 9.61% lower than in 2018 (36,519,951 prescriptions). Similarly, prescriptions per 1000 enrollees in 2019 (464) were 6.47% lower than in 2018 (494). Broad-spectrum antibiotic prescriptions decreased by 14.11% compared to 7.95% for narrow-spectrum agents. Figure 1 shows the quarterly prescriptions. Amoxicillin was the most prescribed antibiotic, accounting for almost half (47.0%) of the total, followed by azithromycin (18.76%), cephalexin (11.80%), trimethoprim-sulfamethoxazole (9.25%), doxycycline (6.42%), ciprofloxacin (4.39%), levofloxacin (1.93%), and moxifloxacin (0.49%). Azithromycin and amoxicillin showed the most dynamic quarterly changes, with the highest levels in the first and fourth quarters (Figure 2).

There was a 3-fold difference between the highest (Kentucky = 975/1000) and lowest prescribing states (Alaska = 325/1000) in 2018 (Figure 3A). Similarly, there was a 2.86-fold difference between the highest (Kentucky = 855/1000) and lowest (Oregon = 299/1000) states in 2019 (Figure 3B). Nationally, there was some evidence for a West–South gradient, with 7 of the 13 western states making up the 10 lowest-prescribing states, Oregon (50th), Washington (49th), Alaska (47th), Colorado (46th), New Mexico (44th), Hawaii (43rd), and California (42nd); and 9 of the 15 southern states making up the 10 highest-prescribing states, Kentucky (1st), Louisiana (2nd), Mississippi (3rd), Tennessee (4th), Alabama (6th), Georgia (8th), Oklahoma (9th), and Virginia (10th), in 2019. Similarly, Alaska (50th), Oregon (49th), Washington (48th), and Colorado (47th) were the lowest-prescribing states in 2018.

Further examination of geographical patterns was made by grouping states by census regions. The South prescribed 28.80% more antibiotics (590/1000) than the West (420/1000) in 2018. Similarly, the South prescribed 35.57% more antibiotics (563/1000) than the West (363/1000) in 2019. From 2018 to 2019, the South decreased by 4.51%, which was smaller than the declines in the Midwest (−11.21%), Northeast (−9.05%), and West (−13.58%, Figure 4).

The correlation between population-corrected antibiotic prescriptions in 2018 is shown in Table 1. The correlation between the total was significantly associated with most individual agents except for moxifloxacin, and this pattern was replicated in 2019. There were strong correlations between levofloxacin and other antibiotics, again, with the exception of moxifloxacin, for both 2018 and 2019. This same pattern of strong associations was also identified for trimethoprim-sulfamethoxazole.

## 4. Discussion

There are at least three key findings from this pharmacoepidemiological report of US MEPS and Medicaid patients. First, there were reductions in the overall dispensing of antibiotics nationally. A national Medicaid reduction of 6.47% in antibiotic prescribing was observed, with 7.95% for narrow-spectrum and 14.11% for broad-spectrum agents from 2018 to 2019. These changes were generally less pronounced than the reductions for azithromycin (−10.5%), doxycycline (−13.6%), amoxicillin (−16.5%), levofloxacin (−17.7%), and trimethoprim-sulfamethoxazole (−21.7%) from 2018 to 2019 in the MEPS. Looking more broadly in MEPS (i.e., 2013 to 2020), outpatient prescribing decreased by 38.7%, with broad-spectrum (−53.7%) agents declining more than narrow-spectrum (−23.5%). These results are quite different to an earlier MEPS report which determined that broad-spectrum antibiotics doubled, with a 2.5-fold increase for those aged 18–49, from 2000 to 2010 [19]. There are potential differences to explore regarding health behaviors of patients insured through commercial insurance and those insured through Medicaid. For example, a study from Massachusetts found that parents insured through Medicaid were more trusting of the information about coughs, colds, and the flu from product advertisements, social media, and other parents, but less trusting than commercially insured parents of the Centers for Disease Control, the Department of Public Health, and their child’s doctor [28]. This difference in trust levels could have influenced prescribing patterns based on which families visited physicians for symptoms of sickness.

Second, there was sizable regional and pronounced state-level variation in antibiotic prescribing. This included a 2.8-fold difference between the highest (Kentucky and Louisiana) and lowest (Alaska and Oregon) states. These findings are consistent with past geographical findings [7,29]. Analysis of QuintilesIMS reports from 2013 determined that children aged <19 from Louisiana and Mississippi received four-fold more azithromycin prescriptions than children from Oregon or Alaska [29]. Similarly, examination of the IQVIA database revealed 2.7-fold more antibiotic prescriptions in 2019 in Mississippi (1193/1000 patients) than Alaska (447/1000 patients) [7]. Kentucky children from the more rural and eastern Knott County received three-fold more antibiotic prescriptions than more urban children from Louisville [30]. Elevations in southern relative to western states are not limited to Medicaid-insured patients [17,31,32]. However, Medicare Part D patients in the South in 2013 had 1623 antibiotics per 1000 patients, which was only 26% more claims than those in the West (1292) [32].

The variation in antibiotic prescribing in the US can be compared with those identified in other countries. A key report determined that antibiotic dispensing rates were significantly elevated relative to the Canadian national average (627 prescriptions/100 K) in Newfoundland and Labrador (921) but significantly lower in British Columbia (BC, 543). Similarly, macrolide dispensing was twice as high in Newfoundland and Labrador (126) vs. BC (58. Antibiotic prescribing was over two-fold higher in Newfoundland and Labrador (1007) relative to BC (426) among children (age < 18) [33]. Further examination within urban areas in BC revealed that Chinese (OR = 0.71) and non-Asian, non-Whites (OR = 0.60) were less likely to fill antibiotic prescriptions than White women. The ethnicity and gender antibiotic differences can be contrasted with other drug classes. South Asian women (OR = 1.46) and men (OR = 1.61) were more likely to fill prescriptions for statins. On the other-hand, Chinese women (OR = 0.29) and men (OR = 0.11) were much less likely to fill prescriptions for antidepressants [34]. An evaluation of primary care antibiotic use in England’s 2017 Clinical Commissioning Groups showed a two-fold regional variation for antibiotics overall but three-fold for broad-spectrum and seven-fold for cephalosporins [35]. A pharmacist in Mozambique, where there is limited enforcement of laws against antibiotic self-medication, reported “Antibiotics are the most sold medicines, so there is no point in working in a pharmacy and not selling antibiotics without (sic) prescription, you end up fired” [36].

Third, and perhaps most novel, was the finding that there were several strong (r > 0.50) associations between state-level use of different antibiotics in 2018, which were generally consistent when examined for the subsequent year. The state in which a patient resides (e.g., West Virginia versus adjacent Virginia) might be anticipated to have only a slight biological or bacteriological impact, but could be highly influential due to the many social determinants of health. One example of this was an analysis published in 2015 which showed counties with higher income and education levels were associated with a lower level of antibiotic prescriptions. Multivariate analysis showed that higher levels of antibiotic prescribing were more likely to be found in areas with higher levels of obesity, an increased number of healthcare providers, and a greater proportion of females [17]. Also of note was that antibiotic prescribing varied based on the quarter in the year. The highest antibiotic prescribing rates occurred during October to March, while prescribing rates fell from April to the end of September. This is consistent with patterns in upper respiratory tract infections, which are often more prevalent during fall or winter months and decrease during warmer months.

Pharmacoequity is a health equity goal to ensure “individuals, regardless of race, ethnicity, and socioeconomic status, have access to the highest-quality medications required to manage their health needs” [37]. Lower physical access to pharmacies (i.e., “pharmacy deserts”) is another impediment to achieving pharmacoequity [37]. Although this pharmacoepidemiological report was completed with an economically homogeneous (i.e., low socioeconomic status) population, the identification of three-fold state-level differences in overall antibiotic prescribing suggests that re-focused efforts may be necessary to achieve pharmacoequity. Fortunately, there is already an extensive body of research to inform re-invigorated antibiotic stewardship programs [38,39]. A systematic review to determine the profile of those mis-prescribing antibiotics in primary care revealed that older age was a significant risk factor in eight of twelve studies. Medical specialty was identified in nine of fifteen studies as increasing the risk for mis-prescribing, particularly among those who work with pediatric populations [39]. Complacency, defined as an attitude that motivates the prescription of antibiotics to fulfill the expectations that professionals believe they have to patients and parents, was identified as a mis-prescribing risk in fourteen of sixteen studies [39]. The healthcare burden, defined as the volume of patients that passed through a doctor’s practice in a work day, was related to mis-prescribing in seven of thirteen reports [39]. Patient perceptions and misunderstandings about when antibiotic treatment is warranted could also be targeted in educational programs. A fascinating report of primary care patients in Baltimore examined what types of messages would most dissuade patients from requesting nonindicated antibiotics. Statements that emphasized the potential harm to the individual patient, like “Taking antibiotics can hurt your body’s natural defenses. This makes it easier for you to get another infection” or “By changing your normal gut bacteria, antibiotics can cause allergies, asthma, and stomach problems”, were most impactful [38]. Antibiotics were the third most common class with unused medications among rural Medicare patients in central Pennsylvania, with the primary explanation being that the patient “did not need all” [40]. This likely also presents another continued opportunity for medication therapy management (MTM) programs serving Medicaid patients.

Antibiotic use has evolved significantly since their production began. Over one hundred individual antibiotics are currently available among six major classes [41]. In contrast with other pharmacotherapies, antibiotics are unique in that the triad of patient, drug, and pathogen affect not only the choice of drug, but the dose and duration as well. Antibiotic choice relates to several factors, including guideline recommendations, patient characteristics, physician preference, available supply, time of year, and bacterial resistance. Most antibiotic use occurs in the outpatient setting, with a staggering 80–90% of the total volume of antibiotic use occurring here [42]. In the outpatient setting, the most common indications for antibiotics include upper respiratory tract infections, urinary tract infections, and skin infections. A 2011 analysis showed that antibiotics are prescribed at 13% of outpatient visits [43]. Patterns of antibacterial prescribing vary by patient age, season, and geographic location. The highest consumption of antibiotics occurs in patients less than two years and greater than 65 years old [44]. In terms of geographic location, the highest level of antibiotic use has been shown to occur in the South, with the lowest rate occurring in the West [45]. Antibiotic usage guidelines for the most common infections are developed through national organizations, regional health systems, and local hospitals. These guidelines focus on diagnostic and treatment recommendations based on clinical trial data, resistance patterns, and stewardship efforts. Patient characteristics that influence antibiotic prescribing include age, allergies, tolerance, adherence, drug interactions, and history of antibiotic treatment. Tolerance and patient preference significantly impact antibiotic choice. Additionally, many patients have in mind what antibiotic they would like to be prescribed before walking into the office [45]. Widespread school and business shutdowns related to the COVID-19 pandemic began in March of 2020. This drastically changed the logistics of everyday life. Without the inevitable exposure to sickness through school, childcare, and work, it was hypothesized that outpatient antibiotic use would decrease throughout 2020. A CDC analysis showed that antibiotic use during this time varied among settings. In the outpatient setting, antibiotic prescribing dropped in 2020 and 2021 compared to 2019 due to the decrease in outpatient healthcare use overall [46].

There are some strengths and limitations to these two complementary national databases. First, the existence of state-level disparities among Medicaid programs is only suggestive of inappropriate antibiotic prescribing, which has been identified previously [10,11,12,13]. Further research is necessary to determine if these state-level disparities persisted, or were exacerbated, during the COVID-19 pandemic. Second, only eight antibiotics were examined for MEPS and eight for Medicaid. However, prior investigations have noted that the top five antibiotics accounted for 87% of all prescriptions [31]. The MEPS and Medicaid databases are based on claims and do not account for antibiotics that were diverted from other sources. Third, this pharmacoepidemiological report extends past population-level studies where the primary unit of analysis was the country [18,24,47,48,49], state [17,29,44,50], or county [17,51]. Although ecological studies are a key to informing antibiotic stewardship efforts, the databases employed [52,53] do not contain information about specific conditions (e.g., upper respiratory infections or urinary tract infections), which would be valuable to characterize in future investigations of the pronounced state-level disparities among Medicaid patients.

## 5. Conclusions

This study provides an overview of antibacterial prescribing practices for US outpatients and in the Medicaid system. Key findings include disparities in antibacterial prescribing rates among different regions across the US, including southeastern states. In conjunction with prior investigations which have characterized the risk factors for heightened prescribing, we are cautiously optimistic that these findings will contribute to renewed antibiotic stewardship efforts.

## Figures and Tables

**Figure 1 pharmacy-12-00046-f001:**
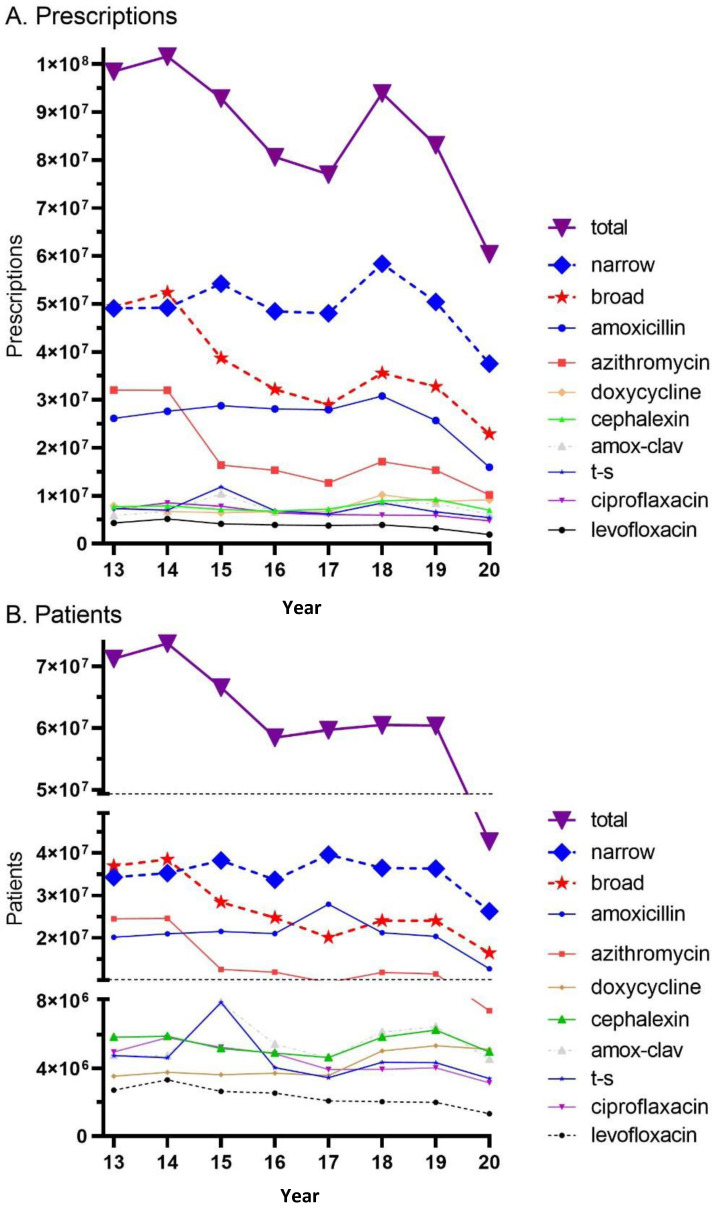
Total prescriptions (**A**) and patients (**B**) prescribed narrow- and broad-spectrum antibiotics, as reported by the Medical Expenditure Panel Survey (MEPS). trimethoprim-sulfamethoxazole (t-s).

**Figure 2 pharmacy-12-00046-f002:**
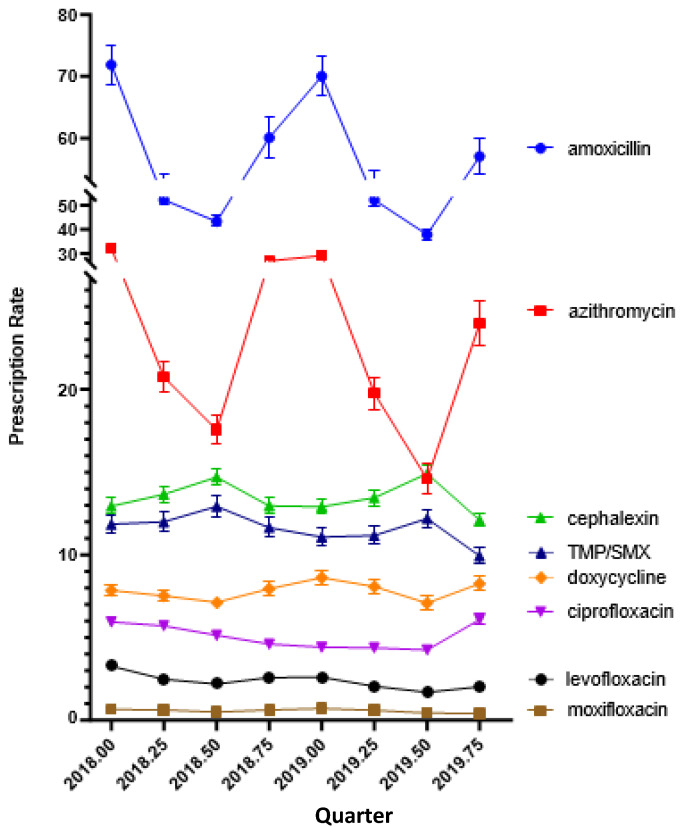
Quarterly antibiotic prescribing per 1000 Medicaid patients in 2018 and 2019.

**Figure 3 pharmacy-12-00046-f003:**
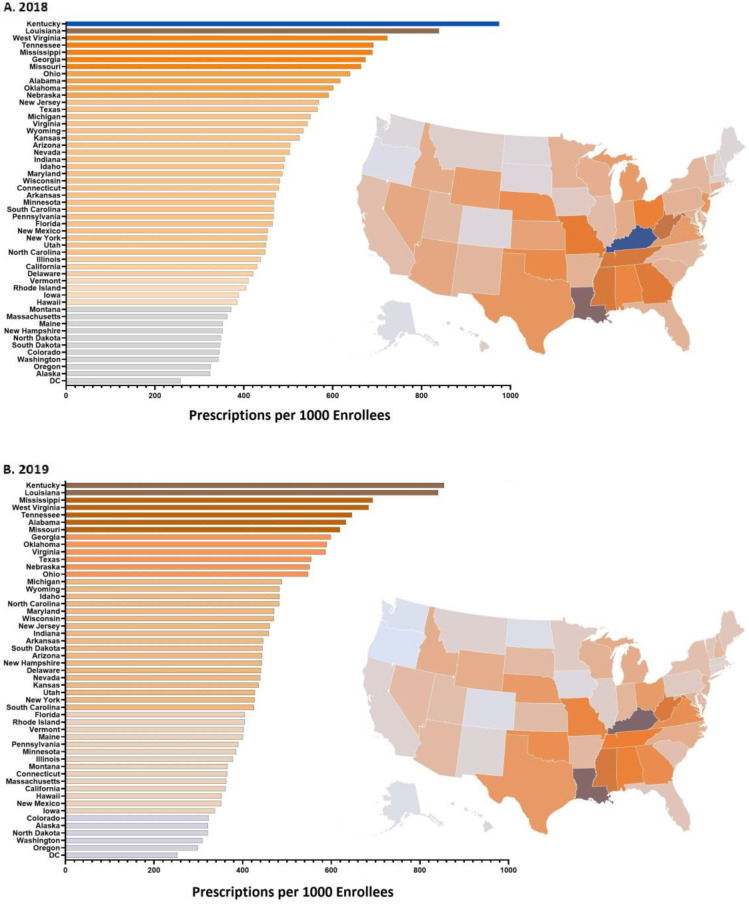
Total antibiotic prescribing rate per state in 2018 (**A**) and 2019 (**B**) per 1000 Medicaid patients.

**Figure 4 pharmacy-12-00046-f004:**
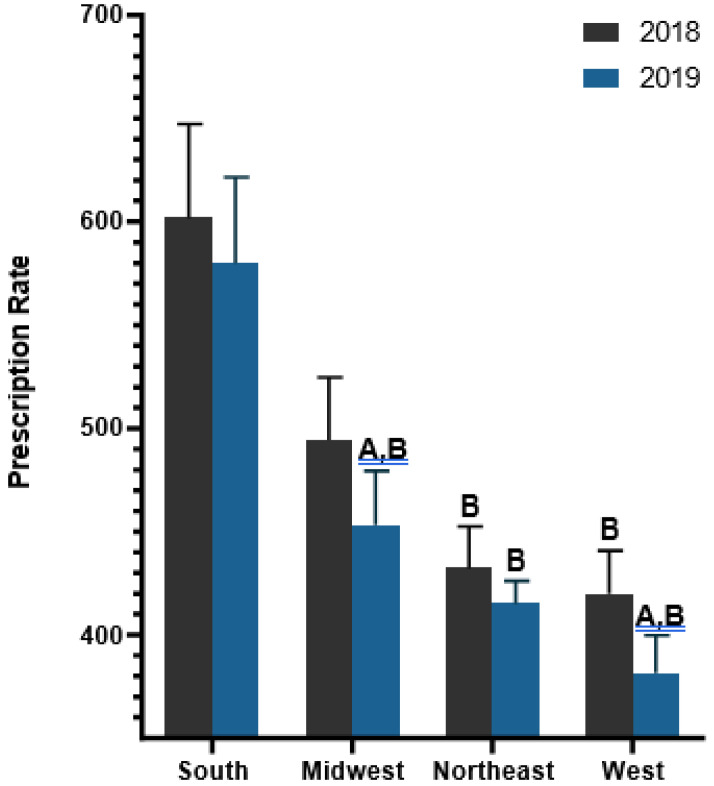
Antibiotic prescribing per 1000 Medicaid enrollees by census regions. ^A^
*p* < 0.05 difference within the region. ^B^ *p* < 0.05 difference compared to the South.

**Table 1 pharmacy-12-00046-t001:** Correlation matrix showing associations (Pearson r) between antibiotic prescribing to Medicaid patients in 2018 (top) and 2019 (bottom). Trimethoprim-sulfamethoxazole: tmp-smx. ^a^
*p* < 0.05 (0.279), ^b^
*p* < 0.01 (0.361).

** 2018 **	** A **	** B **	** C **	** D **	** E **	** F **	** G **	** H **	** Total **
amoxicillin (A)	1.00								
azithromycin (B)	0.81 ^b^	1.00							
cephalexin (C)	0.43 ^b^	0.50 ^b^	1.00						
ciprofloxacin (D)	0.42 ^b^	0.60 ^b^	0.53 ^b^	1.00					
doxycycline (E)	0.14	0.36 ^b^	0.48 ^b^	0.64 ^b^	1.00				
levofloxacin (F)	0.57 ^b^	0.70 ^b^	0.54 ^b^	0.81 ^b^	0.68 ^b^	1.00			
moxifloxacin (G)	0.29 ^b^	0.16	0.03	0.18	0.21	0.07	1.00		
TMP-SMX (H)	0.69 ^b^	0.85 ^b^	0.59 ^b^	0.70 ^b^	0.56 ^b^	0.76 ^b^	0.03	1.00	
total (I)	0.93 ^b^	0.93 ^b^	0.61 ^b^	0.64 ^b^	0.41 ^b^	0.76 ^b^	0.20	0.87 ^b^	1.00
** 2019 **	** A **	** B **	** C **	** D **	** E **	** F **	** G **	** H **	** Total **
amoxicillin (A)	1.00								
azithromycin (B)	0.81 ^b^	1.00							
cephalexin (C)	0.40 ^b^	0.43 ^b^	1.00						
ciprofloxacin (D)	0.61 ^b^	0.76 ^b^	0.47 ^b^	1.00					
doxycycline (E)	0.15	0.30 ^a^	0.41 ^b^	0.59 ^b^	1.00				
levofloxacin (F)	0.57 ^b^	0.67 ^b^	0.52 ^b^	0.85 ^b^	0.64 ^b^	1.00			
moxifloxacin (G)	0.35 ^b^	0.22	0.13	0.03	0.26	0.07	1.00		
TMP-SMX (H)	0.72 ^b^	0.86 ^b^	0.56 ^b^	0.84 ^b^	0.54 ^b^	0.78 ^b^	0.04	1.00	
total (I)	0.94 ^b^	0.92 ^b^	0.55 ^b^	0.79 ^b^	0.39 ^b^	0.75 ^b^	0.23	0.89 ^b^	1.00

## Data Availability

All raw data are publicly available at: https://data.medicaid.gov/, accessed on 17 February 2024.

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
