# Peer review of "Decreases and Pronounced Geographic Variability in Antibiotic Prescribing in Medicaid"

_pharmacy, 2024, doi:10.3390/pharmacy12020046_

Round 1

Reviewer 1 Report

Comments and Suggestions for Authors

I commend the authors for the enormous work they put into this paper. 

The paper is well-written. There are significant aspects that the authors should consider before deeming this paper ready for publication.

Introduction: In the context of the discussed topic, it is not necessary to review the mechanism of action of both broad and narrow-spectrum antibiotics. I recommend that the reviewers remove this information as it doesn't add much value to the paper. I suggest focusing on the geographical variability of antibiotic prescribing. Discuss what has been done in this area, how it has contributed to potential unnecessary antibiotic prescribing, and its risk to public health. Additionally, it shows the audience why understanding these variations in antibiotic prescription across the state is significant.

A gap in the literature needs to be included; discuss the gap in literature you intend to bridge and elaborate on what is novel about your approach or study. Authors stated, “State-level differences in antibiotic prescribing are well-known [18-19]. Geographic areas with high antibiotic consumption have been associated with increased antibiotic resistance, and broad-spectrum antibiotics were shown to be more likely associated with antibiotic resistance.” – then what is novel about your study beyond continuously monitoring antibiotic prescription rates?

Here are two pieces of literature to consider as you develop your study:

  1. https://www.ncbi.nlm.nih.gov/pmc/articles/PMC5840100/
  2. https://www.ncbi.nlm.nih.gov/pmc/articles/PMC9905267/

Methods: It was difficult to follow the design of the study – upon reviewing the method section, I assume that you retrospectively reviewed the quarterly prescription rate from 2013 to 2021, which is then compared across the states in the United States. If this is the case, clearly state that. You may consider having a section titled "Study Design." 

Finally, clearly state the outcome of the study.

Upon reviewing your procedure, I noticed you extracted only the antibiotics used but not the corresponding infectious disease it was treated with.

It would be difficult to understand or interpret what is going on besides the mere interpretation that perhaps there is a high or low level of prescription rate. Such information may not help clinicians. Consider matching based on disease states such as pneumonia, UTI, etc.

Results: See the last paragraph comment.

Discussion: I applaud the authors for clearly discussing their observations. To further strengthen this section, they should discuss the public health implications in the context of their observed findings.

Author Response

Please note that the attached file may be slightly more readable.

1 February, 2024

Dear Pharmacy Editors,

We would like to submit our revision to the manuscript “Decreases and Pronounced Geographic Variability in Antibiotic Prescribing in Medicaid”. We appreciate the reviewers thoughtful and detailed feedback. A point by point response to how their insights (assigned letters) were thoroughly incorporated is below with new text in green highlights and deleted text designated with red strikethrough.

Thank you,

Brian

Reviewer #1

I commend the authors for the enormous work they put into this paper. 

The paper is well-written. There are significant aspects that the authors should consider before deeming this paper ready for publication.

  1. Introduction:In the context of the discussed topic, it is not necessary to review the mechanism of action of both broad and narrow-spectrum antibiotics. I recommend that the reviewers remove this information as it doesn't add much value to the paper.

We have removed the paragraphs about mechanism of action of broad and narrow spectrum antibiotics as requested. This includes:

This study examined four broad spectrum antibiotics (azithromycin, ciprofloxacin, levofloxacin, and moxifloxacin) and four narrow spectrum antibiotics (amoxicillin, cephalexin, doxycycline, and trimethoprim/sulfamethoxazole). Among the four broad spectrum antibiotics, azithromycin was the most commonly prescribed. Azithromycin is a macrolide antibiotic with a long half-life of 68 hours and a high degree of tissue penetration [39]. The drug was initially approved by the FDA in 1991 [39]. Currently, it is primarily used to treat respiratory, enteric, and genitourinary infections and may be used for sexually transmitted infections [39]. The drug is unique to contain additional immunomodulatory effects which makes it a favorable option for chronic respiratory inflammatory diseases [39]. It also differs from other macrolides by having a methyl-substituted nitrogen instead of a carbonyl group at the 9a position on the aglycone ring, which prevents its metabolism [39]. Azithromycin works by binding to the 23S rRNA of the bacterial 50S ribosomal subunit to stop bacterial protein synthesis by inhibiting the transpeptidation/translocation step of protein synthesis and by inhibiting the assembly of the 50S ribosomal subunit which can control of various bacterial infections [39]. The strong affinity of macrolides to bacterial ribosomes, such as azithromycin, is consistent with their broad‐spectrum antibacterial activities [39].

Another broad-spectrum antibiotic is ciprofloxacin which is a second-generation fluoroquinolone that is effective and used only against susceptible bacterial infections including gram-negative and gram-positive bacteria [40]. The drug has a half-life of 4 hours and was approved by the FDA on October 22, 1987 [40]. The mechanism of action of the drug works by inhibiting the bacterial DNA gyrase and topoisomerase IV [40]. Ciprofloxacin binds to bacterial DNA gyrase with 100 times the affinity compared to mammalian DNA gyrase to prevent gyrase supercoiling which inhibits DNA replication [40].

Levofloxacin is a member of the third generation of fluoroquinolones. Along with other quinolones such as moxifloxacin, they are collectively called "respiratory quinolones" due to improved activity against gram-positive bacteria which are commonly implicated in respiratory infections [41]. Like ciprofloxacin, levofloxacin is used to treat susceptible bacterial infections in the upper respiratory tract, skin and skin structures, urinary tract, and prostate, as well as for post-exposure treatment of inhaled anthrax and the plague [41]. The drug was first approved by the FDA in 1996 and has a half-life of 6-8 hours [41]. Similar to ciprofloxacin, the drug works by inhibiting bacterial DNA replication through affecting two key bacterial enzymes: DNA gyrase and topoisomerase IV [41].

The last broad-spectrum antibiotic is moxifloxacin which is a synthetic fluoroquinolone antibiotic agent with a half-life of 11.5-15.6 hours [42]. Moxifloxacin was developed by Bayer AG (initially called BAY 12-8039) and is marketed worldwide under the brand name Avelox (also known as Avalox in some countries) for oral treatment [42]. Currently, the drug is used to treat aerobic gram-positive and gram-negative microorganisms [42]. The drug is well absorbed by the gastrointestinal tract and works by binding and inhibiting DNA gyrase and topoisomerase IV which then inhibits DNA replications [42]. Notably, the drug has 100 times higher affinity for bacterial DNA gyrase than for mammalian [42].

Amoxicillin is the most prescribed antibiotic. The antibiotic is a narrow spectrum penicillin derivative used for the treatment of infections caused by gram-positive bacteria with a half-life of 61.3 minutes [43]. It was first mentioned in literature in 1972 and granted FDA approval on January 18, 1974 [43]. Particularly, amoxicillin is indicated to treat susceptible bacterial infections of the ear, nose, throat, genitourinary tract, skin, skin structure, and lower respiratory tract [43]. The drug works by competitively inhibiting penicillin-binding protein 1 and other high molecular weight penicillin binding proteins, causing bacteriocidal action through inhibiting the ability of bacterial to build and repair their cell wall [43].

The second narrow spectrum antibiotic is cephalexin which is a first generation cephalosporin [44]. This drug has a half-life of 49.5 minutes in a fasting state and 76.5 minutes with food [44]. It was approved by the FDA on January 4, 1971 [44]. The drug works by inhibiting bacterial cell wall synthesis, leading to its breakdown and eventually cell death. Unlike penicillins, cephalosporins are more resistant to the action of beta lactamase [44].

Another narrow spectrum antibiotic is doxycycline which is a tetracycline antibiotic that was first discovered in 1967 and treats a variety of gram-positive and gram-negative bacterial infections [45]. It is also used to treat acne and malaria [45]. The half-life ranges from 12-25 hours [34]. The drug works by inhibiting bacterial protein synthesis by allosterically binding to the 30S prokaryotic ribosomal subunit [45]. Specifically, the drug blocks the association charged aminoacyl-tRNA (aa-tRNA) on the ribosomal A site, which is the acceptor site on the mRNA-ribosome complex to inhibit the elongation phase of protein synthesis which halts the production of essential proteins for bacterial survival and functioning [45]. Doxycycline also mediates anti-inflammatory actions by inhibiting nitric oxide synthase and preventing calcium-dependent microtubular assembly and lymphocytic proliferation to inhibit leukocyte movement during inflammation [45].

The last narrow spectrum antibiotics are trimethoprim/sulfamethoxazole. These two antibiotics are administered in combination with each other and used to treat a variety of infections of the urinary tract, respiratory system, and gastrointestinal tract [47]. The average half-life of sulfamethoxazole is 10 hours and the half-life for trimethoprim ranges from 8-10 hours but both half-lives can be prolonged in patients with severe renal dysfunction [46,47]. These agents work synergistically to block two consecutive steps in the biosynthesis of nucleic acids and proteins which are necessary for bacterial growth and division [46]. Using them in conjunction helps to slow the development of bacterial resistance [46,47].

  1. I suggest focusing on the geographical variability of antibiotic prescribing. Discuss what has been done in this area, how it has contributed to potential unnecessary antibiotic prescribing, and its risk to public health. Additionally, it shows the audience why understanding these variations in antibiotic prescription across the state is significant.

Yes, added a paragraph on geographical variability as requested:

The reasoning behind variations in prescribing rates has also been explored to determine whether the root cause of variation was primarily driven by overuse of antibiotics or varying prescribing patterns. Hersh and colleagues assessed the association between regional differences in the visit rate for selected respiratory conditions and whether antibiotics were warranted based on respiratory condition. In this study, the East South Central region was found to have the highest rate of respiratory conditions resulting in antibiotics, and the Pacific region was found to have the lowest rate. The study found that antibiotic prescribing differences were primarily due to differences in diagnostic approaches, and called for antibiotic stewardship efforts that focus on specifically on accuracy of diagnosis. (Infect Dis Ther. 2018 Mar; 7(1): 171–174.). Another analysis by Bizune and colleagues focused on commercially insured individuals less than 65 years old to determine variability of and factors that influence variability in antibiotic prescribing. Their study also found increased antibiotic prescribing in the South likely due to prescribing pattern differences. The authors called for stewardship efforts that address these differences. (Open Forum Infect Dis. 2023 Feb; 10(2): ofac584.)

  1. A gap in the literature needs to be included; discuss the gap in literature you intend to bridge and elaborate on what is novel about your approach or study. Authors stated, “State-level differences in antibiotic prescribing are well-known [18-19]. Geographic areas with high antibiotic consumption have been associated with increased antibiotic resistance, and broad-spectrum antibiotics were shown to be more likely associated with antibiotic resistance.” – then what is novel about your study beyond continuously monitoring antibiotic prescription rates?

Here are two pieces of literature to consider as you develop your study:

  1. https://www.ncbi.nlm.nih.gov/pmc/articles/PMC5840100/
  2. https://www.ncbi.nlm.nih.gov/pmc/articles/PMC9905267/

Great points. Citation to these key studies has been made as well as more clearly identification of the gaps:

Among commercially insured patients in 2017 with an acute respiratory tract infection, those living in the South were 34% more likely than those in the West to receive an antibiotic [#]. Similarly, examination of medical records from the National Ambulatory Medical Care Survey from 2012 to 2013 revealed that patients in the East South Central region (i.e. AL, KT, MS, TN) were more likely than those in Pacific states (CA, OR, WA) to be diagnosed with a condition where an antiobiotic would be prescribed [#].

...

While regional and state-level disparities in antibiotic prescribing are well-known [2 new suggested citations], it is important to provide updated information to in-form whether stewardship efforts have been successful in driving down these population level differences. The low-income Medicaid population may face challenges that are different from those that are commercially insured.

  1. Methods: It was difficult to follow the design of the study – upon reviewing the method section, I assume that you retrospectively reviewed the quarterly prescription rate from 2013 to 2021, which is then compared across the states in the United States. If this is the case, clearly state that. You may consider having a section titled "Study Design." 

The section was relabeled as “Study design” as suggested. Additional methodological information was added (e.g. “There were two databases (MEPS and Medicaid) for this observational report.” to more clearly set-up the methods.

  1. Finally, clearly state the outcome of the study.

For the methods, added (line 118-9): “The  outcomes were the annual number of prescriptions total, broad vs narrow, and for each agent.” to the MEPS paragraph.

Line 133-4 includes: “The primary outcome was the number of prescriptions (yearly and quarterly) corrected for the number of enrollees.”

For the results, added a big-picture sentence to the MEPS results to clarify the big-picture outcome: “There have been substantial declines in the national pattern of antibiotic prescribing.” Similarly, the Medicaid results section starts with “Examination of prescriptions to Medicaid patients revealed several findings includ-ing three-fold state-level differences.” to better emphasize this outcome and avoid the forest/trees problem.

  1. Upon reviewing your procedure, I noticed you extracted only the antibiotics used but not the corresponding infectious disease it was treated with. It would be difficult to understand or interpret what is going on besides the mere interpretation that perhaps there is a high or low level of prescription rate. Such information may not help clinicians. Consider matching based on disease states such as pneumonia, UTI, etc.

Although we agree that this information would greatly enhance this study, information about the corresponding infectious disease is unavailable in both databases. We have added to the limitations/future directions paragraph: “The MEPS and Medicaid databases employed do not contain information about the cor-responding infectious disease that these agents were used for.”

  1. Results: See the last paragraph comment.

If we had infectious disease data in either database, this would be invaluable information and we would include it.

  1. Discussion: I applaud the authors for clearly discussing their observations. To further strengthen this section, they should discuss the public health implications in the context of their observed findings.

Excellent point! Added a new paragraph: “Although the MEPS … particularly in Southern states and in primary care.”

Reviewer 2 Report

Comments and Suggestions for Authors

Please expand the abstract so it would be clear are you writing about the decline in prescriptions, overall expenditure etc. This part is unclear

Amoxicillin was the predominant antibiotic followed by azithromycin, cephalexin, trimethoprim/sulfamethoxazole, doxycycline, ciprofloxacin, levofloxacin, and moxifloxacin. Substantial geographic variation in antibiotic prescribing existed with 2.8-fold between the highest (Kentucky = 855/1,000) and lowest (Oregon = 299) states

Furthermore, please remove the sentence about correlation as it contains no new data and does not add to abstract. Alternatevly, expand it. Which correlations?

Language correction in lines 33-34

Lines 55-58 are not suitable for introduction, please revise

MEPS was previously used for period up to 2010 but you use it for other year ranges. Please explain

Also provide more explanation what it is. If a survey, provide a sample size calculation or add to limitations that it may not be representative

In fig 1 and 2 describe axis x...

In fig 3, what is on upper and what on lower figure?

Provide explanation for matrix table it is very vague

Author Response

Please note that the attached file may be slightly more readable.

Reviewer # 2

Amoxicillin was the predominant antibiotic followed by azithromycin, cephalexin, trimethoprim/sulfamethoxazole, doxycycline, ciprofloxacin, levofloxacin, and moxifloxacin. Substantial geographic variation in antibiotic prescribing existed with 2.8-fold between the highest (Kentucky = 855/1,000) and lowest (Oregon = 299) states

  1. Furthermore, please remove the sentence about correlation as it contains no new data and does not add to abstract. Alternatevly, expand it. Which correlations?

Expanded as suggested “There were significant correlations across states (r = .81 for azithromycin and amoxicillin).”

  1. Language correction in lines 33-34

We are open to further clarification of what needs adjusting in “Overuse and inappropriate prescribing of antibiotics have been identified in a variety of healthcare settings.”

  1. Lines 55-58 are not suitable for introduction, please revise

This content has been deleted “A recent systematic review of 160 observational studies focused on antibiotic exposure and long-term health outcomes in childhood showed that postnatal antibiotic exposure was associated with asthma (odd ratio = 2.0), wheezing (OR= 1.8), allergic rhino conjunctivitis (OR= 1.7), food allergies (OR=1.4), overweight (OR= 1.2) and obesity (OR = 1.2) [17].”

  1. MEPS was previously used for period up to 2010 but you use it for other year ranges. Please explain

A rational for the years selected is now listed in the methods as requested: “This ten-year period was selected to provide an update on past research [20].”

  1. Also provide more explanation what it is. If a survey, provide a sample size calculation or add to limitations that it may not be representative.

Have included additional information including the very respectable sample size (35,000) (note that 1,004 is considered standard in most national surveys to obtain a modest (3%) margin of error: https://www.scientificamerican.com/article/howcan-a-poll-of-only-100/ , and that it is representative: “MEPS is sponsored by the Agency for Healthcare Research and Quality and is nationally representative of the US civilian non-institutionalized population. The pharmacy component involves verifying patient reported information about prescrip-tion drugs [#].

Citation

[#] Abdus S, Hill SC, Ahrnsbrak R. Outpatient Prescription Drugs: Data Collection and Editing in the 2021 Medical Expenditures Panel System. Methodology Report #37, January 2024.

  1. In fig 1 and 2 describe axis x...

A label (Year) has been added to the X-axis on Fig 1. A label has been added to the X-axis (Quarter) of Fig 2.

  1. In fig 3, what is on upper and what on lower figure?

A label has been added onto the upper (2018) and lower (2019) panels of Fig 3 as requested.

  1. Provide explanation for matrix table it is very vague

Added a bit more information to the legend (Pearson r) as requested. Also added more set-up in the data-analysis section as this measure may be less familiar for this audience: “A positive r value indicated that states that prescribed more of antibiotic A also prescribed more of antibiotic B.”

Round 2

Reviewer 1 Report

Comments and Suggestions for Authors

The authors have made commendable revisions to the paper. However, there are several minor and major issues that still need to be addressed:

Minor:

  1. The statement "Sulfonamides and urinary anti-infective agents are the classes most likely to be prescribed without documentation [11]." is unclear. Does this mean without indication?
  2. The sentence "The outcomes were the annual number of prescriptions total, broad vs narrow, and for 189 each agent." is confusing. Please clarify. Are you referring to the annual number of antibiotic prescriptions across US states?

Major:

  1. There is a significant concern regarding the lack of comparison of antibiotics based on specific disease states. This could be perceived as a methodological flaw in the study. Please refer to the previous review comment (original draft). Without such comparison, it is difficult to confidently assert that "There were significant correlations across states in antibiotic prescribing." To address this concern and strengthen the paper, I strongly recommend that the authors include data comparing antibiotic use across states for a specific condition such as upper respiratory infections or UTIs. This would provide valuable context and enhance the credibility of the findings. Alternatively, if this is not feasible, the authors should acknowledge this limitation in the paper. However, I believe that this is achievable based on the data they have.

Overall, the authors have made significant improvements to the manuscript, but addressing these issues will further enhance the quality and impact of the study.

Comments on the Quality of English Language

None

Round 3

Reviewer 1 Report

Comments and Suggestions for Authors

Not applicable